# Application of SWATH Mass Spectrometry and Machine Learning in the Diagnosis of Inflammatory Bowel Disease Based on the Stool Proteome

**DOI:** 10.3390/biomedicines12020333

**Published:** 2024-02-01

**Authors:** Elmira Shajari, David Gagné, Mandy Malick, Patricia Roy, Jean-François Noël, Hugo Gagnon, Marie A. Brunet, Maxime Delisle, François-Michel Boisvert, Jean-François Beaulieu

**Affiliations:** 1Laboratory of Intestinal Physiopathology, Faculty of Medicine and Health Sciences, Université de Sherbrooke, Sherbrooke, QC J1H 5N4, Canada; 2Centre de Recherche du Centre Hospitalier Universitaire de Sherbrooke, Sherbrooke, QC J1H 5N4, Canada; 3Department of Immunology and Cell Biology, Faculty of Medicine and Health Sciences, Université de Sherbrooke, Sherbrooke, QC J1H 5N4, Canada; 4Allumiqs, 975 Rue Léon-Trépanier, Sherbrooke, QC J1G 5J6, Canada; 5Department of Pediatrics, Faculty of Medicine and Health Sciences, Université de Sherbrooke, Sherbrooke, QC J1H 5N4, Canada; 6Department of Medicine, Faculty of Medicine and Health Sciences, Université de Sherbrooke, Sherbrooke, QC J1H 5N4, Canada

**Keywords:** inflammatory bowel disease, IBD biomarkers, SWATH, DIA mass spectrometry, quantitative proteomics, machine learning, bioinformatics analysis, SVM, data mining

## Abstract

Inflammatory bowel disease (IBD) flare-ups exhibit symptoms that are similar to other diseases and conditions, making diagnosis and treatment complicated. Currently, the gold standard for diagnosing and monitoring IBD is colonoscopy and biopsy, which are invasive and uncomfortable procedures, and the fecal calprotectin test, which is not sufficiently accurate. Therefore, it is necessary to develop an alternative method. In this study, our aim was to provide proof of concept for the application of Sequential Window Acquisition of All Theoretical Mass Spectra-Mass spectrometry (SWATH-MS) and machine learning to develop a non-invasive and accurate predictive model using the stool proteome to distinguish between active IBD patients and symptomatic non-IBD patients. Proteome profiles of 123 samples were obtained and data processing procedures were optimized to select an appropriate pipeline. The differentially abundant analysis identified 48 proteins. Utilizing correlation-based feature selection (Cfs), 7 proteins were selected for proceeding steps. To identify the most appropriate predictive machine learning model, five of the most popular methods, including support vector machines (SVMs), random forests, logistic regression, naive Bayes, and k-nearest neighbors (KNN), were assessed. The generated model was validated by implementing the algorithm on 45 prospective unseen datasets; the results showed a sensitivity of 96% and a specificity of 76%, indicating its performance. In conclusion, this study illustrates the effectiveness of utilizing the stool proteome obtained through SWATH-MS in accurately diagnosing active IBD via a machine learning model.

## 1. Introduction

Inflammatory bowel disease (IBD) is a chronic disorder of the gastrointestinal tract that affects millions of people worldwide. It is characterized by inflammation of the intestinal mucosa, leading to symptoms such as abdominal pain, diarrhea, rectal bleeding, and weight loss [1]. During flare-ups, patients require drug treatment, such as steroids, immunosuppressants, and biological therapies, to reduce inflammation and promote healing [2]. Several other diseases and conditions can present symptoms similar to those of IBD, including celiac disease, irritable bowel syndrome (IBS), and infectious colitis [3]. However, each of these diseases requires different treatments. Consequently, the rapid and accurate diagnosis of IBD flare-ups is essential to ensure appropriate treatment and management of this condition. This is especially true as IBD is associated with both unique and severe complications, sometimes requiring hospitalization and intestinal resection.

Currently, the gold standard for diagnosing and monitoring IBD is colonoscopy and biopsy, invasive procedures that can be uncomfortable and present risks of complications [4]. Moreover, IBD is a lifelong disease, and repeated colonoscopies are necessary for disease follow-up, representing a significant burden for patients. It is therefore necessary to develop non-invasive methods for IBD diagnosis and follow-up [5]. Stool biomarkers have emerged as a promising non-invasive approach for IBD diagnosis and monitoring because they are in direct contact with the affected area of inflammation and pathology in IBD and can be utilized repeatedly as required. Among stool biomarkers, protein biomarkers have several advantages over other molecules since they are more stable in stool samples and can provide information on the activity and severity of the disease. Calprotectin is a calcium-binding protein that is released by inflammatory cells and is highly elevated in the feces of patients with IBD [6]. Calprotectin is a common clinically used fecal biomarker to monitor disease activity and the response to treatment and to distinguish between IBD and other gastrointestinal conditions that may have similar symptoms. However, it is not always accurate, and false-positive or false-negative results can occur. Especially when the calprotectin value falls within the range of 100 to 300 µg/g, it can be challenging to predict the transition from the remission phase to the flare-up phase of IBD [7]. Given this, it is reasonable to expect that combining multiple biomarkers could enhance accuracy and sensitivity in diagnostic or research applications [8].

Recently, there have been promising developments in technology and platforms that can identify and measure a large number of targets simultaneously, such as mass spectrometry-based approaches. Mass spectrometry holds great potential for clinical proteomics, which is used for a comprehensive study of proteins in clinical samples with the aim of discovering the most relevant disease markers [9]. Data-independent acquisition (DIA) mass spectrometry enables comprehensive quantification of all detectable proteins in a sample and allows retrospective data analysis. It also has several advantages over data-dependent acquisition (DDA) for proteomic profiling, such as higher reproducibility, a lower missing value rate, and better quantification accuracy [10]. In comparison to various DIA methods [11], Sequential Window Acquisition of All Theoretical Mass Spectra (SWATH) typically provides a combination of deep proteome coverage capabilities with quantitative consistency and accuracy [11,12,13].

Overall, only a few published studies have used mass spectrometry (MS) analysis on human stool samples to identify protein profiles for specific pathologies, including IBD. For example, a pilot study was conducted on a cohort of 10 to discriminate between active and remission phases. However, they did not use a validation group and identified 30 differentially expressed proteins in two groups of five patients [14]. Another study was performed on a cohort of IBD patients, which utilized a spectrum analysis instead of quantitative data. Their validation cohort yielded low specificity (55%), and the standard operating procedure (SOP) for sample collection and storage in this study required dispatch to the laboratory within 2 h and freezing at −80 °C, which may not be compatible with the general constraints of a standard clinical setup [15]. Recently, Vitali et al. identified three single fecal biomarkers using 2-DIGE and MALDI-TOF/TOF MS on stool samples [16]. Among them, only RhoGDI2 showed better performance than calprotectin to discriminate control from IBD patients. However, this marker, like calprotectin, was not able to identify patients in the middle zone, encompassing those in remission and with moderate activity.

Nevertheless, these studies demonstrated the feasibility of using mass spectrometry on stool samples to identify specific biomarkers that can contribute to the diagnosis of IBD.

Alternatively, analyzing such a large DIA dataset, especially from complex samples such as stool, is challenging and necessitates advanced bioinformatics to identify reliable patterns. In this regard, machine learning (ML) and using advanced feature selection methods have emerged as promising tools. Our hypothesis was that conducting a proteomic analysis on clinical laboratory samples that are intended for the fecal calprotectin test would enable the development of a highly sensitive and specific non-invasive stool test based on mass spectrometry. To investigate this hypothesis, we combined and applied our expertise in basic research, clinical practice, and bioinformatics to develop a precise machine learning model for the accurate diagnosis of active IBD patients from symptomatic non-IBD patients.

This study represents a significant advancement in the field by demonstrating the effectiveness of SWATH-DIA proteomic profiling in diagnosing active IBD patients from non-IBD controls. The novel integration of this proteomic approach with machine learning techniques to create a predictive model enhances the diagnostic accuracy. The model’s practicality was confirmed through successful validation of a separate set of samples, achieving 96% sensitivity with a 0.96 AUC. Furthermore, the robustness of the model is evident in its ability to process data from multiple batches with different collection times, showcasing its real-world applicability. Importantly, the stool samples were obtained under clinically compatible SOP conditions, emphasizing the study’s relevance to clinical practice.

## 2. Materials and Methods

### 2.1. Sample Collection and Research Ethics

A total of 123 samples was obtained from the Clinical Hematology Lab of the CIUSSS de l’Estrie-CHUS in the context of the fecal calprotectin (f-cal) testing program. The research protocol for accessing stool samples from patients that have been tested for f-cal includes a reverse consent procedure for using residual stool samples and accessing the related clinical data on the Ariane network for diagnosis. This protocol has been approved by the Research Ethics Committee of the CIUSSS de l’Estrie-CHUS (Protocol number 1991-17, 90-18; last date of approval 27 August 2023). Patients under 18 years were excluded from the study. When prescribed an f-cal test by their doctor, patients were instructed to collect a stool sample at home and bring it to the hospital within 24 h (according to the CHUS protocol, 2 h max at RT, within 24 h, but in the fridge (4 °C)). In the Hematology lab, a special device was used to collect a fixed amount of stool (~50 mg) and perform the extraction to be tested for calprotectin using ELISA. The remaining stool samples were stored frozen at −80 °C and waited for confirmation of the patient’s lack of objection from the Archive Division before being stored in the lab and included in the study.

Furthermore, in our study, we excluded samples with ambiguous diagnoses, retaining only those with clear-cut diagnoses made using imaging, colonoscopy, fecal calprotectin tests, and histological data by the attending physician. The control group in our study consisted of individuals who consulted a doctor for symptoms mimicking IBD. However, subsequent tests confirmed the absence of IBD in these patients. The control group predominantly consisted of individuals with irritable bowel syndrome (IBS), and some had infectious colitis. Hence, we refer to them as symptomatic non-IBD controls.

### 2.2. Sample Preparation

Sample preparation was implemented as previously described [17]. Briefly, 100 mg of frozen stool specimens was solubilized in 1 mL of lysis buffer (25 mM Tris, 1% SDS, pH 7.5) and centrifuged. Then the aqueous phase between the pellet and the floating residual was recovered and stored at −80 °C until preparation for LC-MS/MS analysis. The concentration of solubilized proteins in the individual samples was measured using a BCA test. For reduction, the samples were treated with 10 mM dithiothreitol (DTT) and, for alkylation, the samples were exposed to 15 mM iodoacetamide. Subsequently, the quenching step was implemented using 10 mM DTT. The proteins were precipitated with cold acetone and methanol and digested with Trypsin/Lys-C. The cleaning and recovery of the peptides were performed with a reverse-phase Strata-X polymeric SPE sorbent column (Phenomenex, Torrance, CA, USA) according to the manufacturer’s instructions. The recovered peptides were dried under nitrogen flow at 37 °C for 45 min and stored at 4 °C until being resuspended in 20 µL of mobile phase solvent A (0.2% *v*/*v* formic acid and 3% DMSO *v*/*v* in water) before LC-MS/MS analysis.

### 2.3. SWATH-MS Data Acquisition

The acquisition of LC-MS/MS data was conducted at the proteomics facility located at Allumiqs Solutions in Sherbrooke, Quebec, Canada. Samples were analyzed using an Eksigent μUHPLC (Eksigent, Redwood City, CA, USA) coupled to an ABSciex TripleTOF 6600 mass spectrometer equipped with an electrospray interface with a 25 μm i.d. capillary. Data-Independent Acquisition (DIA) Sequential Window Acquisition of All Theoretical Mass Spectra (SWATH) acquisition mode was used to acquire raw data from the individual samples. The source voltage was set to 5.5 kV and maintained at 325 °C, the curtain gas was set at 35 psi, gas one was set at 27 psi, and gas two was set at 10 psi. Separation was performed on a reverse-phase Kinetex XB column with a 0.3 mm i.d., 2.6 μm particles, 150 mm (Phenomenex), which was maintained at 60 °C. Samples were injected by loop overfilling into a 5 μL loop. For the 60 min LC gradient, the mobile phase consisted of the following: solvent A (0.2% *v*/*v* formic acid and 3% DMSO *v*/*v* in water) and solvent B (0.2% *v*/*v* formic acid and 3% DMSO in EtOH) at a flow rate of 3 μL/min. DDA analyses were conducted with a 60 min LC gradient, while SWATH analyses utilized a 30 min LC gradient under the following conditions: 0 to 4 min, maintaining a constant 98%/2% solvent A/B mixture; 4 to 16 min, transitioning to a 75%/25% mixture; 16 to 21 min, transitioning to a 55%/45% mixture; 21 to 25 min, transitioning to 100% solvent B, which continued until 27 min; and 27 to 30 min for column re-equilibration. The decision to reduce the LC gradient length to 30 min for SWATH was driven by logistical considerations. To ensure optimal SWATH data quality, various combinations of parameters were assessed using variable acquisition windows for an MS scanning range from 350 to 1250 *m*/*z*. Parameters evaluated encompassed the number, width, and distribution of the SWATH windows, as well as ion accumulation times. Optimization of SWATH windows was executed using the SWATH Variable Window Calculator (Sciex), scaling window sizes across the *m*/*z* range based on the *m*/*z* intensity distribution. The selected optimized SWATH method was determined by identifying the combination that provided a minimum of 6 MS2 data points per peak while maximizing quantifiable proteins and peptides.

### 2.4. Spectral Library Generation

To generate an ion library, extracted proteins from a representative pool of samples (3 IBD and 3 symptomatic non-IBD patients) were separated on a 4–20% polyacrylamide gel and then reduced, alkylated, and digested in the gel. Peptides were extracted from the gel using successive rounds of dehydration and sonication and purified using reverse-phase SPE. Data-Dependent Acquisition (DDA) mode was used to acquire raw data from 12 gel fractions of a pooled sample. The spectral library was created following the procedure outlined in a previous study [17]. Briefly, the raw data (.wiff) files obtained in DDA and DIA mode were converted into mzML format with MSConvert (GUI) from ProteoWizard (v3.0.22074) [18]. Subsequently, we utilized FragPipe software (https://fragpipe.nesvilab.org/, accessed on 10 March 2022) to search the MS/MS spectra against the human proteome reviewed database (UP000005640; including isoforms and contaminants; accessible at www.uniprot.org (accessed on 15 March 2022), containing 20,411 reviewed proteins) via the MSFragger search engine [19]. This search was conducted with default open search parameters, specifying a peptide length between 6 and 42, using strict trypsin as the enzyme with a maximum of 1 missed cleavage allowed, setting the maximum fragment charge to 4, and designating methionine oxidation as a variable modification and carbamidomethylation as a fixed modification. The mass tolerance for precursor ions was set to ±20 ppm and for fragment ions at 20 ppm. The false discovery rate (FDR) for both peptide and protein identifications was set at 5%. The DDA and DIA-based libraries were merged and carefully filtered to remove duplicated precursors and we counted a total of 2000 proteins. This integration increased the human proteome coverage of the library.

### 2.5. Label Free Quantification Analysis

All DIA-converted data in mzML format were processed using DIA-NN software (version 1.8.1) with the following parameters: a fragment ion *m*/*z* range of 200 to 1800, a precursor *m*/*z* range of 300 to 1800, a precursor false-discovery rate (FDR) threshold of 1%, automatic settings for mass accuracy at both the MS2 and MS1 levels, and the scan window. Protein inference was set to ‘Genes’, and the quantification strategy was ‘robust LC (high accuracy)’. Cross-run normalization was disabled, while match between runs (MBR) was enabled.

### 2.6. Statistical and Modeling Analysis

The statistical analysis was conducted with R software (version 4.2.2) and the basement of RStudio included packages ggplot2 for visualization, limma [20] for normalization, sva [21] for batch effect correction, and impute for imputation [22]. Differentially expressed proteins were identified using ProStar software (version 1.30.5) [23]. Machine learning and the feature selection analysis were mainly performed using freely available WEKA software (https://www.cs.waikato.ac.nz/ml/weka/, version 3.8.6, accessed on 15 January 2023) [24] and using R packages Caret (Classification And REgression Training) [25], caretEnsemble [26] and Boruta [27].

The mass spectrometry proteomics data have been deposited to the ProteomeXchange Consortium via the PRIDE [28] partner repository with the dataset identifier PXD047585.

## 3. Results

### 3.1. Patient Demographics

A total of 123 stool samples were collected, including 70 active IBD patients and 53 gastrointestinal symptomatic non-IBD patients. The age distribution of the samples was 48.3 ± 19.8 (mean ± SD) and ranged from 18 to 90 years, and the sex distributions in each group lay approximately in an equal range (53% F vs. 47% M) with no statistical difference.

### 3.2. MS Analysis and Generating the Spectral Library

The samples were analyzed using SWATH-MS in four distinct batches with four replicated samples in batches for the batch effect diagnosis. Initially, we used batches 1–3 including 78 samples for the retrospective analysis and model training, while keeping aside batch 4 with 45 samples as a prospective blind testing group. For accurate peptide identification, we utilized the combined library (DDA and DIA) in conjunction with MBR (match between runs) within the DIA-NN software. The DIA-NN software employs collections of deep neural networks to enhance the ability to match DIA fragmentation patterns with spectral libraries, thereby improving sensitivity [29]. Moreover, enabling the match between runs (MBR) parameter led to an increase in the average number of identified entities and significantly improved data completeness by reducing the occurrence of missing values (https://github.com/vdemichev/DiaNN, accessed on 15 May 2022). An estimated 1250 proteins and 9000 peptides were identified and quantified.

### 3.3. Data Preprocessing

To obtain a precise differential expression protein (DEP), it is necessary to conduct an accurate data analysis of quantitative proteomic studies. This involves various key steps in data processing, including normalization, batch effect correction, imputation of missing values, and appropriate statistical analysis [30,31,32,33]. Since there is currently no established standard procedure for data processing in quantitative proteomics, to ensure an accurate biomarker analysis, we optimized each analytical step and identified an appropriate pipeline, as summarized in Figure 1. To begin the analysis, we first eliminated contaminants and proteins that had less than 70% valid values in each batch. After completing this step, we were left with a total of 250 proteins for further analysis. Afterward, a logarithm transformation (log_2_) was applied to the intensity values as common practice for normalizing skewed data and approximating a normal distribution. To evaluate the data structure initially, we used a box plot to observe differences in variances and means [31,33] (see Figure 2a). From Figure 2a, it is clear that there is significant variation among the samples and batches, which indicates the batch effect. In order to eliminate the unwanted non-biological variability caused by differences in procedures of sample collection, storage, preparation, and spectral data acquisition, a normalization process was necessary [33].

#### 3.3.1. Normalization and Batch Effect Correction

Normalization methods must be selected carefully. Several studies have been carried out on this topic. Dubois et al. systematically evaluated various commonly used normalization methods on a large MS-based proteomic dataset [34]. The results indicated that there was superior performance for certain methods, including sample quantile normalization and median centering. Due to the approximate similarity of sample proteomes, we employed quantile normalization, which is designed to align the distributions of different samples by matching their quantiles [35]. Zhao et al. suggest utilizing a “class-specific” approach for quantile normalization in their study. However, as we intended to apply the final pipeline to an unseen dataset with a blind group label, we opted for overall normalization (regardless of classification) instead [36]. Figure 2b shows the intensity distributions after quantile normalization, displaying high similarity, which is desirable in experiments in which most features are expected to remain constant. Although normalization improves comparability among samples, it primarily focuses on aligning their overall patterns. Consequently, even after normalization, batch effects that specifically impact particular proteins or protein groups can remain a significant source of variance. To explore if data were affected by batches, a principal component analysis (PCA) was performed using batch labels. The results depicted in Figure 2c highlight the considerable influence of the batch on the sample distribution and clustering of samples. Moreover, the replicated samples were generally not closely grouped, except for replicate D, which could randomly position. This clustering can be caused by external experimental factors such as technical and temporal variability. In addition, we attempted to apply median-centering normalization as an alternative to quantile normalization to assess its impact on the batch effect. However, the PCA results did not demonstrate any noticeable improvement with this approach (Appendix A). To remove the batch effect, we used the ComBat method, which is a popular and widely used method for gene expression data but is also applicable to proteomics data [37]. ComBat offers an enhanced variant of the mean shift that makes use of a Bayesian framework. The application of the ComBat algorithm to normalized data yielded a substantial improvement in correcting batch effects, as seen in Figure 2d,e. This improvement was evident in the closest representation of the replicated samples of each batch, as observed in the PCA analysis shown in Figure 2f. Furthermore, even though we were aware that it is preferable to perform batch correction after normalization [38], we wanted to explore if the order in which normalization and batch correction are implemented had any impact on the outcome. However, their PCA comparison indicated there were no significant differences observed between the two approaches (Appendix A).

#### 3.3.2. Missing Value Imputation

Missing values (MVs) are commonly encountered in quantitative proteomics datasets, primarily due to the limitations of protein detection and random fluctuations that occur during the process of data acquisition. The presence of MVs necessitates the consideration of their removal or imputation [33]. To determine the most appropriate approach for handling missing values, it is crucial to identify the origins and types of these missing values [39]. In general, MVs can be categorized into three types: missing values not at random (MNAR), missing values at random (MAR), and missing values completely at random (MCAR) [40]. The analysis of the data from each batch and condition, categorized as symptomatic non-IBD or active IBD, revealed the absence of intentionally missing values. In other words, we did not have proteins that were exclusively present under just one condition. Additionally, comparing replicated samples confirmed the random nature of the MVs. Various studies have assessed different imputation methods to handle missing values [40,41]. However, these studies have yielded varying results in terms of the best method due to the differences in the datasets used. Nevertheless, random forest (RF) [42] and k-nearest neighbors (KNN) [43] are commonly recommended for addressing random missingness [31].

Notably, Wang et al. have introduced the NAguideR toolkit [44], which incorporates 23 commonly used imputation methods and provides evaluation criteria to assist researchers in selecting the most suitable method for their dataset. When we applied this toolkit to our dataset, RF and KNN ranked first and second as the most appropriate imputation methods. After evaluating both methods, we found that neither of them was significantly superior to the other. However, we ultimately decided to proceed with the KNN method, considering N = 5 for the processed data, which means that the KNN method utilizes a machine learning algorithm to estimate missing values based on the values of their five closest neighbors in the feature space.

#### 3.3.3. Identifying Differentially Expressed Proteins (DEPs)

Following data cleaning and preprocessing, protein abundance data were prepared for further statistical analysis and downstream investigation. Our goal was to identify the subset of proteins that demonstrated significant changes between the two conditions among the pool of 250 proteins. The *t*-test and limma [20] are two widely used hypothesis testing methods. In our analysis, we utilized ProStar software [23], which incorporates both of these methods and provides options for both the *t*-tests (Student’s and Welch’s) and limma. Considering our assumption of varying data variation and different sample sizes in the two study groups, we chose to employ Welch’s *t*-test [45]. We applied two criteria via ProStar to identify differentially expressed proteins, a fold change (FC) ratio of at least 1.6 (i.e., |Log_2_(FC)| ≥ 0.70) and a *p*-value less than 0.05 (i.e., Log_10_(*p*-value) ≥ 1.3), resulting in the filtration of 201 proteins [46]. The subsequent *p*-value calibration plot assessed the *p*-value distribution and allowed an FDR estimation adjustment using various statistical methods, such as st.boot, st.spline, langaas, Benjamini–Hochberg, etc. [47]. This calibration plot ensures an evaluation of how well observed *p*-values align with the expected behavior under specific assumptions about the proportion of differentially and non-differentially abundant proteins. In this analysis, the st.boot (Bootstrap) method demonstrated superior performance, yielding a pi0 value of 0.05, indicative of effective control over the false discovery rate (FDR) (below 1%). Using these criteria for the training group, we identified 48 DEPs, as shown in the volcano plot (Figure 3). Detailed data related to the *p*-values and fold changes of these 48 proteins are provided in Appendix A. Compared to symptomatic non-IBD patients, there are 32 proteins presented as upregulated and 16 proteins shown as downregulated.

### 3.4. Functional Enrichment Analysis

KEGG and gene ontology enrichment analyses were performed with ShinyGO (http://bioinformatics.sdstate.edu/go/, accessed on 1 July 2023) in order to gain insights into the functional roles of dysregulated proteins. Gene ontology analysis revealed that these 48 proteins are mainly involved in biological processes related to inflammatory and immune responses, particularly neutrophil- and myeloid cell-related processes, as we expected (Figure 4a), which confirms the upregulation of inflammatory genes in patients with active IBD compared to symptomatic non-IBD patients. Moreover, KEGG pathway enrichment analysis results revealed seven pathways that are significantly affected by DEPs, including the renin–angiotensin system, protein digestion and absorption, mineral absorption, the complement and coagulation cascade, the IL-17 signaling pathway, pancreatic secretion, and Chagas disease. The “renin–angiotensin system (RAS)” pathway is highly enriched and likely plays a significant role in IBD, as illustrated in Figure 4b. Some studies have reported altered levels and activities of RAS components in the inflamed mucosa. These studies suggest that RAS inhibition can have anti-inflammatory effects on IBD. That is why pharmacologically inhibiting the classic RAS pathway using ACE inhibitors and angiotensin II receptor blockers (ARBs) has been a well-established strategy to treat hypertension [48]. The next highest fold enriched pathway includes protein digestion and absorption, reflecting alterations in digestive functions and nutrient absorption in IBD patients [49]. Moreover, the role of the IL-17 pathway in the pathogenesis of IBD and its involvement in inflammatory cytokine production, neutrophil recruitment, and tissue remodeling has been demonstrated [50].

### 3.5. Feature Selection

To construct a predictive model, it is essential to select relevant features while removing redundant and irrelevant ones through feature selection. This process reduces data dimensionality, improves model performance, and reduces overfitting. In this study, “features” refer to the “proteins”, and we aimed to reach a reasonable number of proteins as biomarkers. Two common classic feature selection models are the filter and wrapper methods. The main difference between them is that a filter model selects features based on intrinsic data properties, while a wrapper model involves a learning algorithm in determining feature quality [51]. To identify the most relevant features among the 48 DEPs, we assessed five well-known feature selection methods, including correlation-based feature selection (Cfs), Boruta, information gain, gain ratio, and the wrapper method in WEKA software. Among these methods, the Cfs method demonstrated superior prediction performance compared to the others. Cfs is a filter-based feature selection method that chooses features based on their maximum correlation with the class variable and minimum intercorrelation [52]. As feature reduction offers several benefits, including speeding up algorithm processing time, improving data quality, enhancing algorithm predictive power, and making results more understandable, we aimed to investigate whether we could reduce these 16 proteins without compromising classification performance [33]. To refine our selection, we excluded proteins with less attribute weight, resulting in the elimination of five that had minimal impact on classification performance. Seeking further optimization, we assessed protein–protein correlations among the remaining 11 proteins and removed the ones with a high intercorrelation and lower weight attribute. This iterative process led to a reduction in the number of proteins to seven. Figure 5 illustrates the correlation heatmap among proteins, with the selected ones highlighted. This visualization demonstrates that the selected proteins are primarily chosen from distinct clusters, confirming their low intercorrelation.

These seven proteins included six upregulated proteins and one downregulated protein, as displayed in Table 1. Figure 6 illustrates the enhancement in unsupervised group classification via PCA across three datasets: the original dataset comprising 250 proteins, the dataset following the DEP analysis of 48 proteins, and the dataset featuring the seven proteins selected through the feature selection process. Based on these plots, it is visually evident how effectively the two groups separate as we reduce the number of proteins, and the cumulative proportion of variance explained by the first two principal components significantly increases from 22% to 61%.

### 3.6. Selecting the Appropriate Machine Learning Algorithm

Machine learning (ML) is a powerful tool in bioinformatic analysis. Supervised machine learning refers to using quantitative proteome data with known clinical conditions to train a model for the prediction of prospective samples [53]. To identify the most appropriate predictive classifier according to the nature of the data, the five most popular machine learning methods, including support vector machines (SVMs), random forests (RFs), logistic regression (LR), k-nearest neighbors (KNN) and naive Bayes (NB), were evaluated. Table 2 displays the performance metrics of five classifiers for predicting active IBD patients from symptomatic non-IBD patients in terms of accuracy, precision, recall, F-score, area under the ROC curve (AU-ROC), and area under the precision and recall curve (AU-PRC). Detailed information on each of these parameters is presented in Table 3. These metrics were obtained using WEKA [24]. The results indicate that the SVM classifier outperforms the others in all criteria.

### 3.7. Optimizing the Selected Model Performance

In the context of machine learning, there is a risk of a model becoming overly proficient at learning from the training data, a phenomenon known as overfitting. This entails not only capturing the inherent patterns but also incorporating noise or random variations present in the data. Consequently, an overfit model performs exceptionally well on the training data but faces challenges when attempting to apply its knowledge to new and unfamiliar data. Two strategies that help to avoid overfitting are cross-validation and hyperparameter tuning [54]. In this study we used 10-fold cross-validation for the training data where the data divided into ten subsets, with nine parts of the data used for training and one part for validation in each fold. The experiment was then repeated 10 times, with each of these subsets serving as the validation group. The final result indicated the average across the 10 folds, which provides a more realistic assessment of the model’s performance. Moreover, most machine learning algorithms have parameters that can be adjusted, referred to as hyperparameters. These hyperparameters are critical for building robust and accurate models, as they help find the balance between bias and variance, thereby preventing the model from overfitting. Two common effective techniques for hyperparameter tuning are grid search and random search [55]. Rafael et al. have demonstrated equal predictive performance for grid and random search techniques in SVM hyperparameter tuning [56]. In this analysis, we employed the “tuneLength = 10” function for each classifier as a grid search method. This means that the system will perform hyperparameter tuning by randomly selecting 10 different combinations of hyperparameters for each classifier and evaluating their performance using cross-validation to find the best hyperparameter settings. The analysis indicates that, for an SVM classifier with a polynomial kernel of degree 1, a scale parameter of 0.001, and a cost parameter of eight (C = 8), it outperforms other configurations and achieves an accuracy of 0.95, a sensitivity of 0.93, and a specificity of 0.97 in classifying the training dataset.

### 3.8. Model Validation with Prospective Data

To validate the optimized model, it was applied to 40 blind samples from batch 4. The prediction results indicated 96% sensitivity and 76% specificity, as shown in the confusion matrix in Figure 7. Figure 7 also illustrates that the area under the ROC curve was equal to 0.96. The results highlight the high accuracy and performance of the generated model in accurately classifying the blind data.

## 4. Discussion

This study demonstrated the potential use of SWATH-DIA proteomic profiling of stool samples as a tool for diagnosing active-IBD patients from symptomatic non-IBD patients. This was achieved by employing machine learning techniques to develop a robust predictive model. To accomplish this, we designed an experiment with three main steps of (1) data acquisition and processing, (2) training and optimizing a machine learning model based on 78 retrospective samples, and (3) validating the model’s performance on 45 prospective samples. Achieving 96% sensitivity with a 0.96 AUC using a blind dataset confirmed the model’s robustness, also indicating our ability to successfully and effectively process the data obtained from four separate batches with different collection times. The processing steps included the successful removal of batch effects and employed effective methods for normalization and missing value imputation.

We have corrected the batch effect using the ComBat method. ComBat starts by adjusting each batch of data separately to have similar means and variances, and then calculates the differences between the batches and uses this information to “harmonize” the data. ComBat adjusts the data for each sample in a way that minimizes the batch-related differences while preserving the true biological differences [37].

To impute missing values, it is crucial to understand the nature of the data and determine the reasons for their absence, which will guide the selection of an appropriate imputation method. Upon comparing the replicated samples, we observed that the missing values were missing at random. “Zero”, “mean”, and “minimum value” are the straightforward imputation methods commonly used, but they may not always be suitable, especially when the missing values occur randomly and are not due to limits of detection or actual missing data. In such cases, imputing them with these methods could introduce bias into the analysis. Therefore, we chose to employ the k-nearest neighbors (KNN) imputation method with a setting of five neighbors. This implies that it leverages information from the five most similar samples in the dataset to estimate the missing values.

Among the differentially expressed proteins (DEPs), the highest-scoring proteins in the volcano plot were S100A8 and S100A9, which are well-known neutrophil-derived proteins predominantly found as the S100A8/S100A9 complex, also known as calprotectin. This finding further confirms the correctness of the analysis pathway.

Utilizing all 48 differentially expressed proteins as biomarker signatures for classification may not be practical. Therefore, we needed to reduce the number of biomarkers without compromising prediction accuracy. However, selecting only the best proteins and combining them based on previous studies does not guarantee an improvement in overall classification performance. Furthermore, in machine learning, a specific coefficient is assigned to each biomarker, known as a weight, based on its importance and effect on classification to achieve an optimal result. For instance, Mooiweer et al. found that the combination of fecal hemoglobin and calprotectin did not enhance their predictive accuracy compared to using fecal Hb and FC individually [57]. Similarly, Schröder et al. found that the combination of calprotectin, lactoferrin, and neutrophile elastase did not increase predictive accuracy when compared with calprotectin alone [58]. In this regard, using correlation-based feature selection in this study helped us to only keep the seven most relevant proteins with maximum correlations with the class variable and minimum intercorrelation. For instance, retaining both the S100A9 and S100A8 proteins does not provide significant additional informative value because both of them are subunits of calprotectin and exhibit a high correlation with each other. Moreover, the correlation heatmap in Figure 5 indicates that S100A9 and S100A8 also share a high correlation with lactoferrin, and there is also a noticeable correlation between azurocidin, myeloblastin, and myeloperoxidase. Although all of them were identified previously as potential IBD markers, keeping one of them would give us almost similar results.

The seven selected proteins include the upregulated proteins S100A9, azurocidin (AZU1), immunoglobulin lambda constant 3, hemoglobin subunit delta, phospholipase B-like 1 (PLBD1), and alpha-1-acid glycoprotein 1 (alpha 1-AGP), and the downregulated protein neutral ceramidase (ASAH2). Two of these proteins, S100A9 and AZU1, are associated with neutrophils and play a key role in the host’s defense against bacterial infections. S100A9 is, in fact, a subunit of calprotectin, accounting for approximately 60% of the total soluble proteins in the cytosol fraction of neutrophils, while AZU1 is found in the azurophilic granules of neutrophils, alongside other proteins [59]. Hemoglobin delta is linked to occult intestinal bleeding in IBD patients, and previous research has highlighted a correlation between fecal hemoglobin and calprotectin [57]. Immunoglobulin lambda is a light chain of hemoglobin and can be indicative of an active immune system in IBD patients. The increase in free light chains (FLCs), including kappa and lambda immunoglobulins, in plasma has previously been shown in diabetes and immune system abnormalities, as well as autoimmune-based inflammatory diseases [60,61]. However, the dysregulation of lambda light chains in stool and its relevance to IBD have not been studied in detail. PLBD1 is a phospholipase that can generate lipid mediators of inflammation and was first identified in neutrophils [62]. However, to the best of our knowledge, its relationship with IBD has not been specifically investigated. Alpha 1-AGP is one of the major acute phase proteins in humans, and its serum concentration increases in response to systemic tissue injury, inflammation, or infection [63]. Takashi et al. demonstrated a significant increase in fecal alpha 1-AGP in active IBD patients compared to non-active patients, suggesting alpha 1-AGP as a potential biomarker for evaluating IBD activity [64]. ASAH2 is involved in breaking down ceramides to sphingosines. Its downregulation in IBD causes ceramide accumulation in microdomains of cholesterol- and sphingolipid-enriched membranes, resulting in an impairment of the barrier function of the gut [65,66]. The loss of ASAH2 causes elevated levels of sphingosine-1-phosphate and systemic inflammation in ASAH2 knockout mice [67]. These proteins collectively offer insights into the complex molecular mechanisms and potential biomarkers associated with IBD.

The superiority of SVM over other models can be attributed to various factors, including the characteristics of the data, the nature of the classes, the distribution of the features, and the inherent strengths and weaknesses of each algorithm [68]. Some advantages of SVM over other classifiers include being less prone to overfitting due to its optimization process and regularization (controlled by the parameter C and gamma), and greater robustness to outliers and noisy data [69,70].

SVM serves as a robust technique for constructing a classifier [71]. Its primary objective is to establish a decision boundary between two classes, facilitating the classification of data points based on their features. This decision boundary, referred to as a hyperplane, is positioned in a manner that maximizes its distance from the nearest data points of each class, which are known as support vectors [72]. Vapnik initially introduced the SVM algorithm in 1963 to create linear classifiers [73]. Additionally, SVMs can employ kernel methods to model complex, non-linear patterns in higher dimensions. The choice of a suitable kernel function, among other considerations, can significantly impact the performance of an SVM model. However, there is no definitive method to determine the optimal kernel for a specific pattern recognition problem. It often involves a trial-and-error approach, beginning with a basic SVM and experimenting with various standard kernel functions [72]. In this study, the selection of the optimal kernel function is part of the hyperparameter tuning process. Depending on the nature of the data, one kernel (with a degree of one) outperforms the others. This configuration is commonly referred to as “Linear SVM” or “SVM with a Linear Kernel” [74]. This setup assumes that the data is linearly separable, which could be considered an advantage in simplifying the model complexity.

Let us take a closer look at the cost and gamma hyperparameters to gain insights into their impacts on the model. The scale parameter (γ or gamma) controls how tightly the SVM model fits the training data. The usual range for the gamma parameter is typically between 0.01 and 10. Opting for smaller values, such as our chosen value of 0.001, implies a more extensive decision boundary. In contrast, larger values like one or 10 result in narrower decision boundaries, which, if not carefully considered, can potentially trigger overfitting. On the other hand, the cost parameter (C) in SVM controls the trade-off between training error and testing error. The usual range for the cost parameter typically lies between 0.1 and 1000. A smaller C allows for a larger margin and tolerates some misclassification of training points. In our dataset, C = 8 exhibited better performance than the other values. This value strikes a balance between being not too large, which could lead to overfitting, and not too small, which could risk underfitting.

One limitation of this study is that it involved Canadian IBD patients aged 18 and above. Therefore, applying the machine learning algorithm to populations from different regions and ages should be approached with caution. Additionally, while we were able to correct the batch effect, it is essential to note that all samples were analyzed using a single mass spectrometer. To ensure the broader applicability of this method in different clinical laboratories, it would be advantageous to analyze data from various spectrometers.

The primary objective of this study was to provide the proof of concept that a SWATH-based MS analysis can be advantageously used as an additional tool for assisting the gastroenterologist through a protein signature. This, in turn, can significantly enhance the effectiveness of IBD therapy and overall disease management. Moreover, this approach offers substantial advantages in terms of expediting and improving the precision of IBD diagnoses, thereby preventing the deterioration of the patient’s condition due to delayed colonoscopy or inaccurate diagnosis. It also ensures the optimal prescription of drugs from the outset, maximizing treatment efficacy. Additionally, by reducing the necessity for unnecessary colonoscopies, it not only carries financial benefits but also minimizes patient discomfort and anxiety, saves time, enhances convenience, and streamlines the diagnosis and monitoring processes.

In conclusion, this study presents a proof of concept for the application of SWATH for precise IBD diagnosis using stool proteomics and showcases the effectiveness of the data processing and machine learning approaches. Additionally, it highlights the potential of this method for classifying Crohn’s disease (CD) vs. ulcerative colitis (UC) and distinguishing active IBD from remission. The creation of a non-invasive, precise, and sensitive method for diagnosing and monitoring IBD could have a substantial positive impact on the quality of life of IBD patients and lessen the burden of unnecessary or repeated invasive procedures.

## Figures and Tables

**Figure 1 biomedicines-12-00333-f001:**
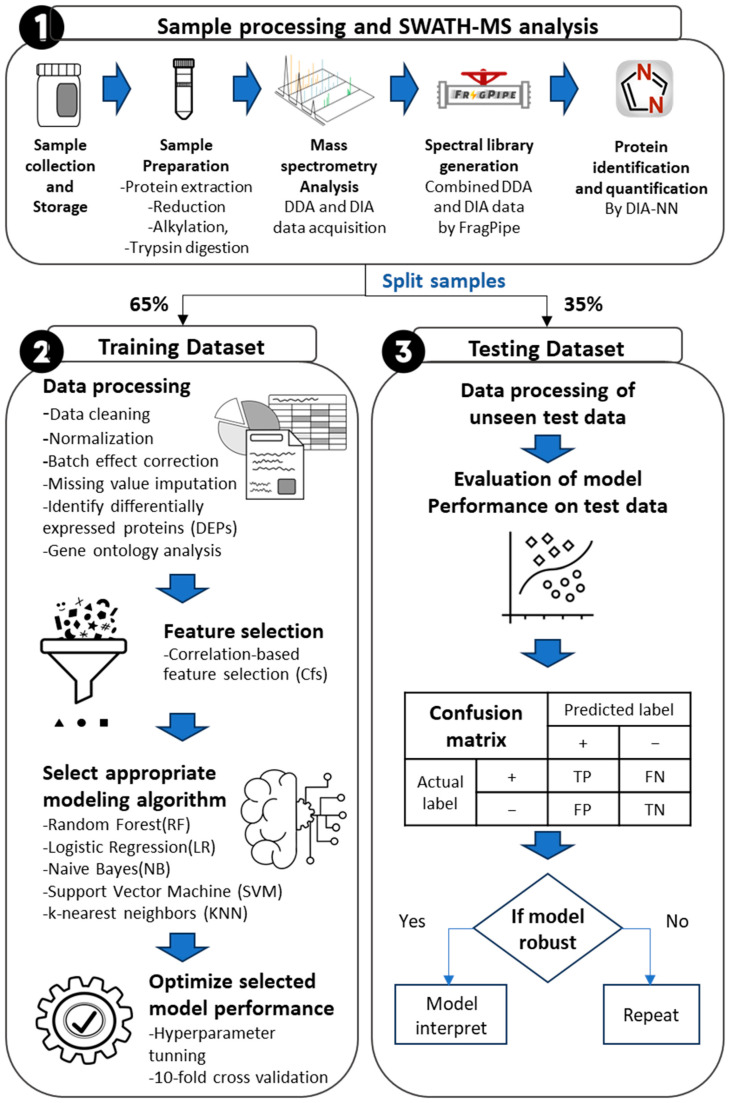
General workflow. This schematic representation outlines the experimental procedure, which consists of three main steps: (**1**) sample processing and SWATH-MS analysis—This step involves obtaining proteome data from stool samples. (**2**) Data processing, training, and optimizing the machine learning model—in this phase, a machine learning model is trained and optimized using 78 training samples. (**3**) Evaluation of model performance—the final step involves evaluating the model’s performance using 45 prospective samples (testing set).

**Figure 2 biomedicines-12-00333-f002:**
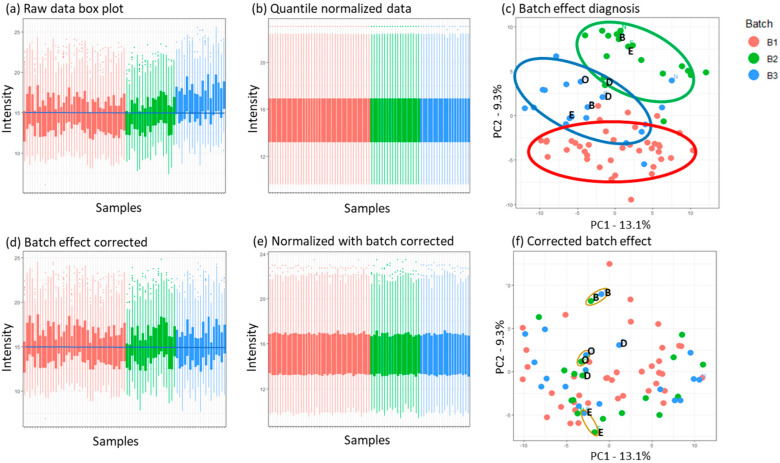
Identification and correction of the batch effect. (**a**) Box plot illustrating the protein distribution in the unprocessed log-transformed data format across three sample batches. (**b**) Box plot displaying the quantile normalized data. (**c**) PCA analysis of the normalized data, revealing clear clustering due to batch effects. (**d**) Box plot representing the influence of the ComBat batch effect correction on the initial data. (**e**) Box plot of data after batch correction of normalized data. (**f**) PCA analysis indicating the successful elimination of batch effects by the close representation of the replicated samples in different batches. The letters in panels (**c**,**f**) illustrate the replicated samples in different batches, which are expected to be seen in close proximity to each other. This expectation is fulfilled after batch correction.

**Figure 3 biomedicines-12-00333-f003:**
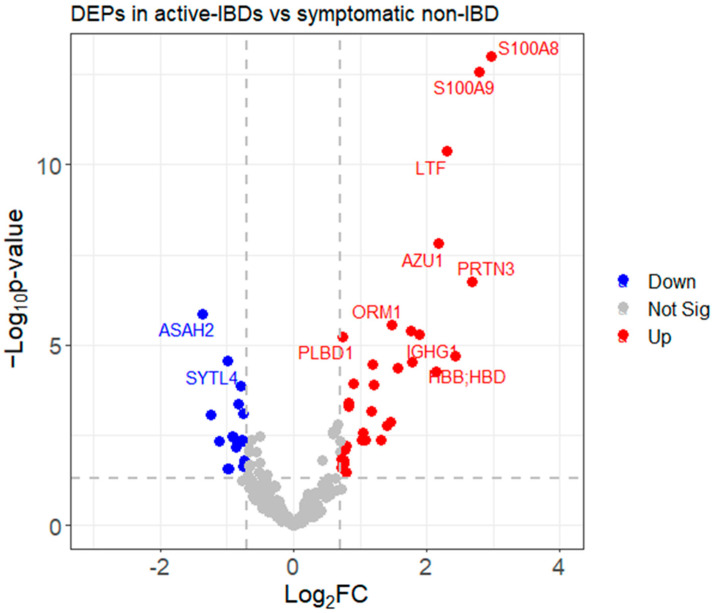
The volcano plot summarizes differentially expressed proteins (DEPs) detected in samples obtained from active IBD patients and symptomatic non-IBD patients. Among the approximately 300 proteins detected as being consistently present in samples obtained from IBD patients, 48 were initially identified either as reduced (blue dots) or increased (red dots). Group comparisons between active IBD patients and symptomatic non-IBD patients were calculated using Welch’s *t*-test with a Log_2_(FC)| ≥ 0.70 and *p*-value ≥ 1.3 in ProStar.

**Figure 4 biomedicines-12-00333-f004:**
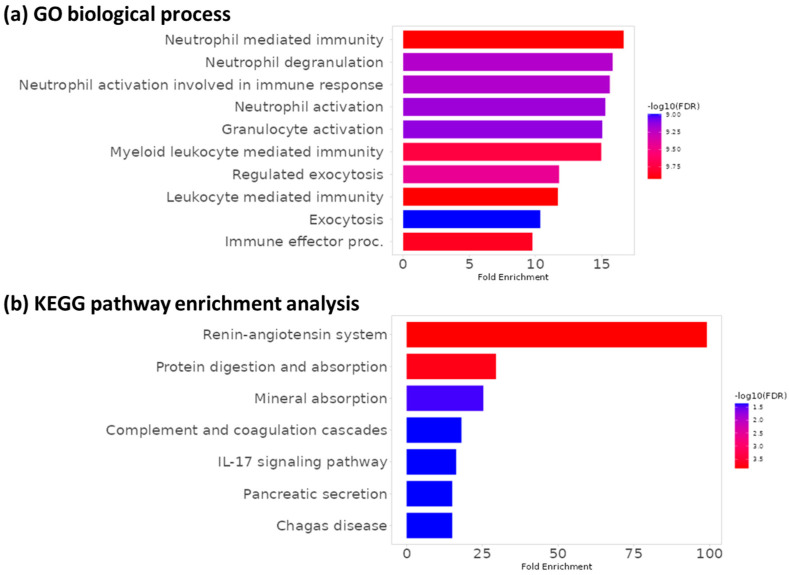
Functional analysis of differentially expressed proteins in the two groups. (**a**) The significant gene ontology analysis of 48 DEPs proteins (*p*-value < 0.05) (**b**) KEGG pathway enrichment analysis of 48 DEPs proteins in active IBD patients and asymptomatic non-IBD patients.

**Figure 5 biomedicines-12-00333-f005:**
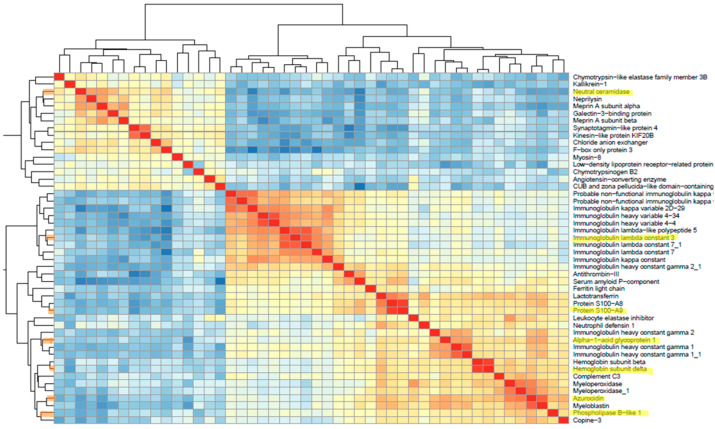
Protein correlation heatmap. This heatmap reveals two prominent clusters: one comprising upregulated proteins and the other containing downregulated proteins. It visually represents correlation strength, with stronger correlations depicted in red and weaker ones in blue. Notably, the seven selected proteins for our model are highlighted in the map, each of which is mainly associated with distinct clusters exhibiting low intercorrelations.

**Figure 6 biomedicines-12-00333-f006:**
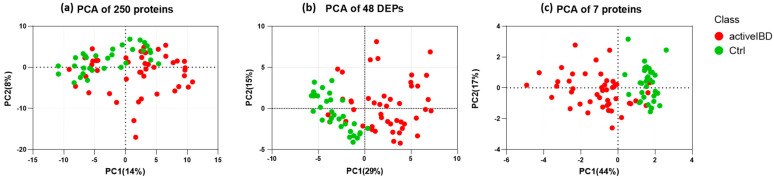
Unsupervised group classification via principal component analysis (PCA) across three datasets: (**a**) the original dataset comprising 250 proteins, (**b**) the dataset following the DEP analysis of 48 proteins, and (**c**) the dataset featuring the seven proteins selected through the feature selection process.

**Figure 7 biomedicines-12-00333-f007:**
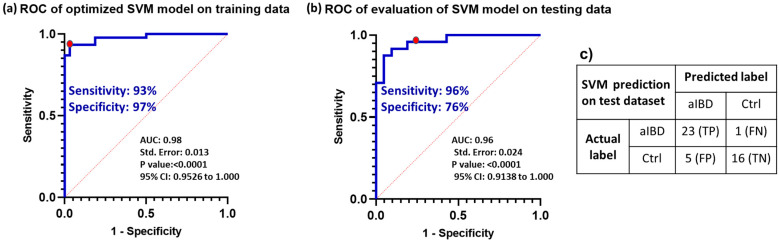
(**a**) The ROC curve analysis involved applying an optimized SVM model to the training dataset to determine the threshold, resulting in 93% sensitivity and 97% specificity as indicated by the red dot. (**b**) In the ROC curve analysis of the model applied to the testing dataset, we obtained an AUC of 0.96. Using a previously selected threshold, the model achieved 96% sensitivity and 76% specificity (red dot). (**c**) The confusion matrix displays the counts of true positives (TP), false positives (FP), true negatives (TN), and false negatives (FN). The results highlight the high accuracy and exceptional performance of the generated model in accurately classifying the blind data.

**Table 1 biomedicines-12-00333-t001:** Characteristics of selected proteins. The list of the final seven selected proteins for a prediction model, including their fold changes and *p*-values, across the two groups. The attribute weights show different levels of importance to different features (proteins) during the model training and classification process.

Protein Name	Gene	Fold Change	*p*-Value	Attribute Weight
Protein S100-A9	S100A9	6.9	0.0000	−3.9813
Azurocidin	AZU1	4.5	0.0000	−2.7925
Immunoglobulin lambda constant 3	IGLC3	2.0	0.0044	−2.4284
Hemoglobin subunit delta	HBB	5.4	0.0000	−2.2529
Phospholipase B-like 1	PLBD1	1.7	0.0000	−1.6708
Alpha-1-acid glycoprotein 1	ORM1	2.8	0.0000	−0.9056
Neutral ceramidase	ASAH2	−2.6	0.0000	1.1675

**Table 2 biomedicines-12-00333-t002:** Performance metrics of classifiers. This table compares the performance metrics of each classifier based on the prediction results obtained from the training and validation of 78 samples using 10-fold cross-validation. The SVM model outperforms the other classifiers based on the first four metrics. However, when considering the area under the curve (AUC), the RF classifier outperforms the others. Further analysis confirms that the SVM model works better for these data and provides more accurate predictions for blind data.

Classifier	Accuracy	Precision	Recall	F-Score	AU-ROC	AU-PRC
SVM	95%	0.97	0.93	0.96	0.95	0.96
NB	90%	0.94	0.90	0.92	0.93	0.94
LR	88%	0.89	0.91	0.90	0.92	0.92
KNN	88%	0.91	0.89	0.90	0.90	0.89
RF	87%	0.89	0.89	0.89	0.94	0.93

**Table 3 biomedicines-12-00333-t003:** General definitions of different performance metrics for model selection.

Measure	Evaluation Focus
Accuracy	The overall effectiveness of a classifier
Precision	The proportion of positive instances among all instances classified as positive
Recall (Sensitivity)	The proportion of positive instances correctly classified as positive out of all positive instances in the data
F-score	The harmonic mean of precision and recall and provides a combined measure of both
ROC Area	The area under the receiver operating characteristic (ROC) curve, which is a graphical representation of the trade-off between the true positive rate and the false positive rate for different threshold values of the classification model
PRC Area	The area under the precision–recall curve (PRC), which is a graphical representation of the trade-off between precision and recall for different threshold values of the classification model

## Data Availability

The mass spectrometry proteomics data have been deposited to the ProteomeXchange Consortium via the PRIDE partner repository (http://www.ebi.ac.uk/pride, accessed 6 December 2023) with the dataset identifier PXD047585.

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
