# Peer review of "Application of SWATH Mass Spectrometry and Machine Learning in the Diagnosis of Inflammatory Bowel Disease Based on the Stool Proteome"

_biomedicines, 2024, doi:10.3390/biomedicines12020333_

Round 1
Reviewer 1 Report
Comments and Suggestions for Authors
The study “Application of SWATH Mass Spectrometry and Machine Learning in Diagnosis of Inflammatory Bowel Disease Based on Stool Proteome” by Elmira Shajari et al. reports that data acquisition of stool proteomic analysis by SWATH and machine learning allowed the development of a robust predictive model to distinguish active IBD from symptomatic non-IBD. The study is interesting and reports new potential biomarkers for IBD diagnosis, a hot topic in medical research.
1. The authors selected 123 stool samples in the context of f-cal testing program. The selection criteria are not clear. Moreover f-cal results of the two groups are not reported. It should be reasonable to compare groups with or without active IBD with overlapping f-cal values. Otherwise a selection bias might occurs. The finding of S100A8 and S100A9 as the most significant differently expressed proteins might reflect this possible bias.
2. To generate an ion library, the authors prepared two pools, one from IBD and another from non-IBD patients, each pool comprising samples from three patients. Which was the criteria for selection of patients entering the pool?
3. Quantitative data obtained by an independent method should be provided to support the finding of the selected 7 proteins.
Author Response
The authors want to thank you for your time and constructive remarks.
Comments and Suggestions for Authors
The study “Application of SWATH Mass Spectrometry and Machine Learning in Diagnosis of Inflammatory Bowel Disease Based on Stool Proteome” by Elmira Shajari et al. reports that data acquisition of stool proteomic analysis by SWATH and machine learning allowed the development of a robust predictive model to distinguish active IBD from symptomatic non-IBD. The study is interesting and reports new potential biomarkers for IBD diagnosis, a hot topic in medical research.
- The authors selected 123 stool samples in the context of f-cal testing program. The selection criteria are not clear. Moreover f-cal results of the two groups are not reported. It should be reasonable to compare groups with or without active IBD with overlapping f-cal values. Otherwise, a selection bias might occur. The finding of S100A8 and S100A9 as the most significant differently expressed proteins might reflect this possible bias.
RESPONSE:
In this study, active-IBD diagnoses were made using imaging, colonoscopy, fecal calprotectin tests, and histological data by the attending physician. According to the protocol, calprotectin levels above 300 µg/mg indicate active disease. As shown in the scatter plot for the calprotectin test, due to the mentioned thresholds, among the 78 training samples, only 19 out of the 46 active patients exceeded this threshold, and the activity of the rest remained ambiguous. However, based on our model we have a clear cut off for discriminating two groups.

|
Figure 1 : A scatter plot illustrates the accuracy of both the calprotectin test and the SVM model in classifying control (ctrl) and active-IBD samples, utilizing predetermined cut-offs. |
Moreover, it is essential to note that the primary objective of this study was not to demonstrate the superiority of our approach over the fecal calprotectin test. Instead, our focus was on showcasing the practical application of mass spectrometry data in stool proteomics for diagnostic testing. In subsequent studies, we will analyze the obtained results in detail, focusing on all subgroup classifications for clinical applications.
- To generate an ion library, the authors prepared two pools, one from IBD and another from non-IBD patients, each pool comprising samples from three patients. Which was the criteria for selection of patients entering the pool?
RESPONSE:
In our study, we excluded samples with ambiguous diagnoses, retaining only those with clear-cut diagnoses. To create ion library pools, we randomly selected samples from patients with confirmed active IBD for the active IBD pool and from non-IBD patients for the control pool. This approach ensures clarity and representation in our ion library.
This information was added to the Sample collection section in the Materials and Methods (lines 140-142).
- Quantitative data obtained by an independent method should be provided to support the finding of the selected 7 proteins.
RESPONSE:
All relevant documents related to the raw files of the samples are accessible on the Pride website as indicated (lines 680-683). Furthermore, the following information has been integrated into the article to enhance clarity (lines 413-430):
To identify the most relevant features among the 48 DEPs, we assessed five well-known feature selection methods, including correlation-based feature selection (Cfs), Boruta, information gain, gain ratio and the wrapper method in WEKA software. Among these methods, the Cfs method demonstrated superior prediction performance compared to the others. Cfs is a filter-based feature selection method that chooses features based on their maximum correlation with the class variable and minimum intercorrelation. As feature reduction offers several benefits, including speeding up algorithm processing time, improving data quality, enhancing algorithm predictive power, and making results more understandable, we aimed to investigate whether we could reduce these 16 proteins without compromising classification performance. To refine our selection, we excluded proteins with less attribute weight, resulting in the elimination of 5 that had minimal impact on classification performance. Seeking further optimization, we assessed protein-protein correlations among the remaining 11 proteins and removed the ones with high intercorrelation and lower weight attribute. This iterative process led to a reduction in the number of proteins to 7. Figure 5 illustrates the correlation heatmap among proteins, with the selected ones highlighted. This visualization demonstrates that the selected proteins are primarily chosen from distinct clusters, confirming their low intercorrelation.
Reviewer 2 Report
Comments and Suggestions for Authors
An in vitro noninvasive predictive model using fecal proteins to differentiate between active IBD and symptomatic non-IBD patients was developed by applying SWATH-MS and machine learning methods in the manuscript. The model identified 48 proteins using Gene Ontology Enrichment Analysis, and to reduce redundancy seven features were selected to train and optimize the machine learning model using a feature selection method for CFs. Ultimately, the model was validated on an unvalidated dataset, the results showed a sensitivity of 96% and a specificity of 76%, indicating its performance. I believe this manuscript will provide an important research basis for researchers working in this field. However, the authors are advised to address the following issues:
(1) There seems to be a discrepancy between the data description and experimental logic used in the division of the training and validation sets in Figure 1. in the text, the ten-fold cross-validation in Chapter 3.7, and the ROC curves of the SVM model in Chapter 3.8? The authors are requested to double-check this to ensure the correctness of the experiments and the accuracy of the results.
(2) Selection of 7 proteins from 48 differential proteins is a very important part of the study, could the authors please answer in detail how the selection was made based on the Protein Correlation Heatmap?
(3) There are some grammatical errors in the article, and the authors are advised to review them carefully and correct them as needed.
(4) The authors are advised to make the data publicly accessible, which not only helps other researchers to further analyze and validate the findings, but also enhances the credibility of the study.
(5) The authors discuss the application of machine learning methods for in vitro disease prediction. Therefore, the authors are advised to cite some bioinformatics articles in the manuscript to enhance the scientific basis of the study, for example, 10.1109/TBDATA.2023.3334673, 10.1186/s12859-023-05309-w and 10.1093/bib/bbac498.
Comments on the Quality of English LanguageMinor editing of English language required
Author Response
The authors want to thank you for your time and constructive remarks.
Comments and Suggestions for Authors
An in vitro noninvasive predictive model using fecal proteins to differentiate between active IBD and symptomatic non-IBD patients was developed by applying SWATH-MS and machine learning methods in the manuscript. The model identified 48 proteins using Gene Ontology Enrichment Analysis, and to reduce redundancy seven features were selected to train and optimize the machine learning model using a feature selection method for CFs. Ultimately, the model was validated on an unvalidated dataset, the results showed a sensitivity of 96% and a specificity of 76%, indicating its performance. I believe this manuscript will provide an important research basis for researchers working in this field. However, the authors are advised to address the following issues:
- There seems to be a discrepancy between the data description and experimental logic used in the division of the training and validation sets in Figure 1. in the text, the ten-fold cross-validation in Chapter 3.7, and the ROC curves of the SVM model in Chapter 3.8? The authors are requested to double-check this to ensure the correctness of the experiments and the accuracy of the results.
RESPONSE:
Thank you for your meticulous review and for highlighting the oversight. I've corrected the error on line 244, where I mistakenly labeled the testing set as the validation set. Just to clarify, in our study, we set aside approximately 35% of the data for blind testing, and the analysis for training and validation of the classifying model was done on the remaining 65%, which includes 78 samples. Given our sample size, we opted for the cross-validation method to validate the developed model. The first ROC curve in Figure 7 illustrates the optimized model applied to the training dataset, while the second ROC curve represents the application of the same model to the testing dataset.
(2) Selection of 7 proteins from 48 differential proteins is a very important part of the study, could the authors please answer in detail how the selection was made based on the Protein Correlation Heatmap?
RESPONSE:
The following information has been integrated into the article to enhance clarity (lines 413-430):
To identify the most relevant features among the 48 DEPs, we assessed five well-known feature selection methods, including correlation-based feature selection (Cfs), Boruta, information gain, gain ratio and the wrapper method in WEKA software. Among these methods, the Cfs method demonstrated superior prediction performance compared to the others. Cfs is a filter-based feature selection method that chooses features based on their maximum correlation with the class variable and minimum intercorrelation [53]. As feature reduction offers several benefits, including speeding up algorithm processing time, improving data quality, enhancing algorithm predictive power, and making results more understandable, we aimed to investigate whether we could reduce these 16 proteins without compromising classification performance [34]. To refine our selection, we excluded proteins with less attribute weight, resulting in the elimination of 5 that had minimal impact on classification performance. Seeking further optimization, we assessed protein-protein correlations among the remaining 11 proteins and removed the ones with high intercorrelation and lower weight attribute. This iterative process led to a reduction in the number of proteins to 7. Figure 5 illustrates the correlation heatmap among proteins, with the selected ones highlighted. This visualization demonstrates that the selected proteins are primarily chosen from distinct clusters, confirming their low intercorrelation.
(3) There are some grammatical errors in the article, and the authors are advised to review them carefully and correct them as needed.
RESPONSE:
Thank you for your feedback. We have carefully reviewed and corrected them in the final version.
(4) The authors are advised to make the data publicly accessible, which not only helps other researchers to further analyze and validate the findings, but also enhances the credibility of the study.
RESPONSE:
All relevant documents related to the raw files of mass spectrometry analysis of samples are accessible on the Pride website as indicated (lines 680-683) in the “Data Availability Statement” section at the end of the manuscript.
(5) The authors discuss the application of machine learning methods for in vitro disease prediction. Therefore, the authors are advised to cite some bioinformatics articles in the manuscript to enhance the scientific basis of the study, for example, 10.1109/TBDATA.2023.3334673, 10.1186/s12859-023-05309-w and 10.1093/bib/bbac498.
RESPONSE:
Thank you for your valuable suggestions. While we appreciate the recommendation of the articles you provided, upon careful review, we found that our current manuscript already cites relevant bioinformatics articles that contribute to the scientific foundation of our study. However, we acknowledge the importance of the suggested articles and will consider them for future research and references in upcoming papers.
Minor editing of English language required.
RESPONSE:
English revision has been done.
Reviewer 3 Report
Comments and Suggestions for Authors
The manuscript proposed by Shajari and co-workers (biomedicines-2788569) entitled "Application of SWATH Mass Spectrometry and Machine Learning in Diagnosis of Inflammatory Bowel Disease Based on Stool Proteome" describes the application of SWATH and machine learning in 108 identifying a protein signature from stool samples obtained under clinical compatible SOP 109 conditions for an accurate diagnosis of active IBD from symptomatic non-IBD patients. The manuscript is well-prepared, however, before publication in Biomedicines, some changes, explanations, and comments are needed.
My major comments are presented below.
1. The novelty and applicability of the performed study should be better exposed. Please make changes.
2. Materials and methods section - what was the control group? What are the parameters of the control group?
3. Materials and methods section - what was the time of sample storage before the final analysis? Is there a significant effect of sample storage on the observed results?
4. Materials and methods section - how were the MS and LC-MS parameters optimized?
5. Materials and methods section - page 4, lines 153-154 - For the 60 min LC gradient, the mobile phase consisted of the following: solvent A (0.2% v/v formic acid and 3% DMSO v/v in water) and solvent B (0.2% v/v for-154 mic acid and 3% DMSO in EtOH) at a flow rate of 3 μL/min. - what was the gradient and how the gradient was optimized? Is the column affecting the obtained results analyzed?
- The quality of Figure 2 is low. The same is for Figure 5.
What is the reproducibility of the proposed method?
How many times was one sample analyzed?
What about the difficulties during the sample preparation and analysis?
Check and correct English.
Check and correct the reference list.
Comments on the Quality of English LanguageSome English corrections (grammar, style) are needed.
Author Response
The authors want to thank you for your time and constructive remarks.
Comments and Suggestions for Authors
The manuscript proposed by Shajari and co-workers (biomedicines-2788569) entitled "Application of SWATH Mass Spectrometry and Machine Learning in Diagnosis of Inflammatory Bowel Disease Based on Stool Proteome" describes the application of SWATH and machine learning in 108 identifying a protein signature from stool samples obtained under clinical compatible SOP 109 conditions for an accurate diagnosis of active IBD from symptomatic non-IBD patients. The manuscript is well-prepared, however, before publication in Biomedicines, some changes, explanations, and comments are needed.
My major comments are presented below.
- The novelty and applicability of the performed study should be better exposed. Please make changes.
RESPONSE:
This paragraph has been added to the end of introduction (lines 110-118):
This study represents a significant advancement in the field by demonstrating the effectiveness of SWATH-DIA proteomic profiling in diagnosing active-IBD from non-IBD controls. The novel integration of this proteomic approach with machine learning techniques to create a predictive model enhances the diagnostic accuracy. The model's practicality was confirmed through successful validation on a separate set of samples, achieving 96% sensitivity with a 0.96 AUC. Furthermore, the robustness of the model is evident in its ability to process data from multiple batches with different collection times, showcasing its real-world applicability. Importantly, the stool samples were obtained under clinically compatible SOP conditions, emphasizing the study's relevance to clinical practice.
- Materials and methods section - what was the control group? What are the parameters of the control group?
RESPONSE:
This paragraph has been added to the Material and Method to enhance clarity (lines 140-146):
Furthermore, in our study, we excluded samples with ambiguous diagnoses, retaining only those with clear-cut diagnoses made using imaging, colonoscopy, fecal calprotectin tests, and histological data by the attending physician. The control group in our study consisted of individuals who consulted a doctor for symptoms mimicking IBD. However, subsequent tests confirmed the absence of IBD in these patients. The control group predominantly consisted of individuals with irritable bowel syndrome (IBS), and some had infectious colitis. Hence, we refer to them as symptomatic non-IBD controls.
- Materials and methods section - what was the time of sample storage before the final analysis? Is there a significant effect of sample storage on the observed results?
RESPONSE:
The samples underwent storage for a period of 6 months to 1 year before analysis. Previous studies have shown that stool samples maintain stability when stored at -80 degrees Celsius. Notably, our observations did not reveal any significant effects associated with the storage time on the obtained results.
- Materials and methods section - how were the MS and LC-MS parameters optimized?
RESPONSE:
The answer to this question is provided in response to the following question.
- Materials and methods section - page 4, lines 153-154 - For the 60 min LC gradient, the mobile phase consisted of the following: solvent A (0.2% v/v formic acid and 3% DMSO v/v in water) and solvent B (0.2% v/v for-154 mic acid and 3% DMSO in EtOH) at a flow rate of 3 μL/min. - what was the gradient and how the gradient was optimized? Is the column affecting the obtained results analyzed?
RESPONSE:
The following information has been integrated into the article to enhance clarity (lines 173-189):
The LC mobile phase consisted of solvent A (0.2% v/v formic acid and 3% DMSO v/v in water) and solvent B (0.2% v/v formic acid and 3% DMSO in EtOH) at a flow rate of 3 μL/min. DDA analyses were conducted with a 60-minute LC gradient, while SWATH analyses utilized a 30-minute LC gradient, under the following conditions: 0 to 4 min, maintaining a constant 98%/2% solvent A/B mixture; 4 to 16 min, transitioning to a 75%/25% mixture; 16 to 21 min, transitioning to a 55%/45% mixture; 21 to 25 min, transitioning to 100% solvent B, which continued until 27 min; and 27 to 30 min for column re-equilibration. The decision to reduce the LC gradient length to 30 minutes for SWATH was driven by logistical considerations. To ensure optimal SWATH data quality, various combinations of parameters were assessed using variable acquisition windows for an MS scanning range from 350 to 1250 m/z. Parameters evaluated encompassed the number, width, and distribution of the SWATH windows, as well as ion accumulation times. Optimization of SWATH windows was executed using the SWATH Variable Window Calculator (Sciex), scaling window sizes across the m/z range based on m/z intensity distribution. The selected optimized SWATH method was determined by identifying the combination that provided a minimum of 6 MS2 data points per peak while maximizing quantifiable proteins and peptides.
- The quality of Figure 2 is low. The same is for Figure 5.
RESPONSE:
High-quality versions of Figure 2 and Figure 5 have been provided in separate files.
What is the reproducibility of the proposed method?
RESPONSE:
We assessed the reproducibility of our method by analyzing samples across four separate batches at different times. Additionally, to evaluate the predictive performance, one batch was kept as a blind test. Employing batch effect correction methods, specifically the combat method, contributed to the consistency of our results. After thorough data processing and application of the prediction model, our findings demonstrate that our method exhibits favorable reproducibility.
How many times was one sample analyzed?
RESPONSE:
Typically, each sample was analyzed once. However, we incorporated some samples as replicates across different batches for a more comprehensive assessment.
What about the difficulties during the sample preparation and analysis?
RESPONSE:
We did not encounter any difficulties during the sample preparation and analysis process.
Check and correct English.
RESPONSE:
Done.
Check and correct the reference list.
RESPONSE:
Done according to MDPI.
Reviewer 4 Report
Comments and Suggestions for Authors
The manuscript discusses the potential use of expression proteomics of stool samples as a tool for discriminating active IBD from symptomatic non-IBD patients. Overall, the manuscript is well-written, easy to read, and worthy of publication. However, there are concerns that need addressing before acceptance:
- Section 3.3.3: Identifying Differentially Expressed Proteins (DEP) What type of p-value correction was applied (e.g., Benjamini-Hochberg correction)? In my opinion, using p < 0.05 and a fold change (FC) ratio of at least 1.6 is not a sufficient condition. Due to the multiple testing problem, it's necessary to clarify how the false discovery rate (FDR) is estimated here. The authors should provide clarification on these points.
- I missed a table detailing the concentration of the 48 differentially expressed proteins in each individual sample. Providing this table would be beneficial for readers interested in performing additional statistical analyses.
Minor comments:
- Line 89: Use a minus sign instead of a hyphen (-80°C).
- Line 601: "The following supporting information can be downloaded at: 00000." This should be corrected.
Author Response
The authors want to thank you for your time and constructive remarks.
Comments and Suggestions for Authors
The manuscript discusses the potential use of expression proteomics of stool samples as a tool for discriminating active IBD from symptomatic non-IBD patients. Overall, the manuscript is well-written, easy to read, and worthy of publication. However, there are concerns that need addressing before acceptance:
Section 3.3.3: Identifying Differentially Expressed Proteins (DEP) What type of p-value correction was applied (e.g., Benjamini-Hochberg correction)? In my opinion, using p < 0.05 and a fold change (FC) ratio of at least 1.6 is not a sufficient condition. Due to the multiple testing problem, it's necessary to clarify how the false discovery rate (FDR) is estimated here. The authors should provide clarification on these points.
RESPONSE:
The following information has been integrated into the article to enhance clarity (lines 356-366):
We applied two criteria via ProStar software to identify differentially expressed proteins: a fold change (FC) ratio of at least 1.6 (i.e., |Log2(FC)| ≥ 0.70) and a p-value less than 0.05 (i.e., Log10(p-value) ≥ 1.3)., resulting in the filtration of 201 proteins (Wieczorek, S., et al., Protein-level statistical analysis of quantitative label-free proteomics data with ProStaR. Proteomics for Biomarker Discovery: Methods and Protocols, 2019: p. 225-246). The subsequent p-value calibration plot assessed p-value distribution and allowed for the adjustment of FDR estimation using various statistical methods, such as st.boot, st.spline, langaas, Benjamini-Hochberg, etc. (Giai Gianetto, Q., et al., Calibration plot for proteomics: A graphical tool to visually check the assumptions underlying FDR control in quantitative experiments. Proteomics, 2016. 16(1): p. 29-32.). The calibration plot ensures an evaluation of how well observed p-values align with expected behavior under specific assumptions about the proportion of differentially and non-differentially abundant proteins. In this analysis, the st.boot (Bootstrap) method demonstrated superior performance, yielding a pi0 value of 0.06, indicative of effective control over the false discovery rate (FDR) (below 1%).
This comprehensive approach effectively addresses the challenges associated with multiple testing problems, providing robust statistical support for the identified DEPs.
I missed a table detailing the concentration of the 48 differentially expressed proteins in each individual sample. Providing this table would be beneficial for readers interested in performing additional statistical analyses.
RESPONSE:
All relevant documents related to the raw files of the samples are accessible on the Pride website as indicated (lines 680-683) in the “Data Availability Statement” section at the end of the manuscript. We also provide the list of 48 proteins with their p-value and fold change in an excel file as supplementary Table1.
Minor comments:
Line 89: Use a minus sign instead of a hyphen (-80°C).
RESPONSE:
Corrected.
Line 601: "The following supporting information can be downloaded at: 00000." This should be corrected.
RESPONSE:
The editor will change it after article submission.
Round 2
Reviewer 3 Report
Comments and Suggestions for Authors
The revised version of the manuscript entitled entitled "Application of SWATH Mass Spectrometry and Machine Learning in Diagnosis of Inflammatory Bowel Disease Based on Stool Proteome" proposed by Shajari and co-workers (biomedicines-2788569) meets my requirements. The authors presented comments and answers to all of my questions and doubts.
Comments on the Quality of English LanguageMinor editing of English language required